# Bridging Bag-Level and Instance-Level Uncertainty with Conformalizable MIL

## Abstract

This paper introduces the concept of Conformalizable Multiple Instance Learning as well as a theoretical framework that establishes the connection between agnostic PAC learnability and the transferability of bag-level conformal prediction guarantees to individual instances. Our analysis defines rigorous conditions under which a calibrated conformal threshold provides reliable uncertainty quantification at both the bag and instance levels. We demonstrate that instance-level agnostic PAC learnability is both a necessary and sufficient condition to achieve valid instance-level coverage. Empirical evaluations on synthetic CIFAR-based tasks, Camelyon16 whole-slide images, and time series anomaly detection task validate our theoretical findings, confirming that agnostic PAC learnability underpins the conformalizability of existing MIL models. This work provides a robust theoretical and empirical foundation for integrating conformal prediction into MIL, offering valuable insights for enhancing uncertainty quantification in complex learning scenarios.

## 1 Introduction

Multiple Instance Learning (MIL) is a powerful weakly-supervised paradigm for learning from ambiguously labeled data, where models learn from bags of instances to make predictions about the instances within them (Dietterich et al., 1997; Andrews et al., 2002; Stikic et al., 2011; Ilse et al., 2018; Shao et al., 2021). This approach has seen wide success, particularly in domains like computational pathology and document analysis where instance-level labels are prohibitively expensive to acquire (Ilse et al., 2018; Li et al., 2021; Shao et al., 2021; Javed et al., 2022; Angelidis & Lapata, 2018). While modern MIL models often achieve high accuracy at the bag level, a critical challenge remains: reliably quantifying the prediction uncertainty for each individual instance. Without trustworthy instance-level uncertainty, deploying these models in high-stakes applications like medical diagnosis or scientific discovery is fraught with risk.

Standard uncertainty quantification methods like split conformal prediction can provide rigorous coverage guarantees at the bag level (Vovk et al., 2005), but it is unclear if these guarantees can be transferred to the instance level without instance labels. This raises a crucial question: *Can bag-level supervision provide reliable instance-level uncertainty under certain conditions?*

This paper provides a theoretical and empirical answer to this question. We introduce the concept of a **Conformalizable MIL**, a model where a conformal threshold $\tau_\alpha$ calibrated on bag-level data can be provably transferred to provide valid instance-level coverage. Our core contribution is to establish the formal link between this practical property of conformalizability and the theoretical property of **agnostic PAC learnability**.

Recent work by Jang & Kwon (2025a) provided a foundational theory for when MIL models are learnable at the instance level from a PAC-theoretic perspective. Our work builds directly upon this, asking the crucial next question: *if a model is instance-level learnable, what does that imply for the reliability of its uncertainty estimates?* Our contribution is therefore not just an analysis, but a prescriptive framework: by connecting learnability to conformalization, we provide a guide for designing and selecting MIL architectures that are not only accurate but also yield trustworthy, calibrated uncertainty estimates at the fine-grained instance level.

Our main contributions are:

- We introduce the **Conformalizable MIL** framework and prove that instance-level agnostic PAC learnability is the **necessary and sufficient condition** for a MIL model to be conformalizable.

- We derive a **finite-sample coverage bound** for instance-level predictions, showing that the coverage guarantee degrades gracefully with the model's excess generalization risk.

- We provide extensive **empirical validation experiments** on independent, dependency-driven, and ordered time-series datasets, demonstrating that our theory correctly predicts which architectures successfully transfer uncertainty guarantees in complex settings.

Ultimately, our work provides a robust theoretical and empirical foundation for integrating conformal prediction into MIL, offering a principled path toward enhancing uncertainty quantification in complex, weakly-supervised learning scenarios.

## 2 CONFORMALIZING THE MIL MODELS

### 2.1 PRELIMINARIES

We begin by considering the basic building blocks of our framework, starting with the essential concepts of MIL. In MIL, rather than treating every single data point on its own, we group related data points into what we call bags. Each bag is composed of several individual elements, known as instances, and each bag is assigned a label that identifies its class. This arrangement allows us to capture richer information by considering collections of instances together.

**Definition 1** (Instance, Bag, and Label). Let $\mathcal{X}_{\text{inst}} \subseteq \mathbb{R}^d$ denote the feature space of an instance. A *bag* is defined as $\mathcal{X} = \left\{ x_{\text{inst}}^i \in \mathcal{X}_{\text{inst}} \mid i = 1, \ldots, N \right\}$, a permutation-invariant collection of $N$ instances. The label space is $\mathcal{Y} = \{y_1, \ldots, y_K\}$. The *bag domain* $D_{\mathcal{X} \times \mathcal{Y}}$ is the joint distribution over bags and their bag-level labels, while the *instance domain* $D_{\mathcal{X}_{\text{inst}} \times \mathcal{Y}}$ governs instances paired with individual labels. Note that the bag domain is the joint probability distribution composed of multiple instance domains.

In simpler terms, while each instance is a point in a high-dimensional space, a bag is an unordered collection of these points. The overall classification task, whether it's distinguishing between two classes or among several, is then approached by considering the relationships and patterns within these bags rather than isolated instances.

The goal of MIL is to learn a classifier that can accurately assign labels to entire bags, based on the information provided by the instances within each bag. In practice, we are provided with a training dataset of labeled bags, and our task is to combine the predictions from individual instance classifiers to determine the bag-level prediction. This is achieved by integrating an appropriate pooling function that aggregates the instance-level outputs into a single bag-level decision.

**Definition 2** (MIL problem). Given a training set $S = \{(\mathbf{x}_m, \mathbf{y}_m)\}_{m=1}^M$ of $M$ bag-label pairs, where each bag $\mathbf{x}_m = \{x_{m,i}\}_{i=1}^{N_m}$ contains $N_m$ instances ($x_{m,i} \in \mathcal{X}_{\text{inst}}$). Each $(\mathbf{x}_m, y_m)$ is drawn IID (Independent and Identically Distributed) from the bag domain $D_{\mathcal{X} \times \mathcal{Y}}$. MIL learns a bag classifier $f_{\text{bag}} : \mathcal{X} \to \mathcal{Y}$, typically $f_{\text{bag}}(\mathbf{x}) = \sigma(f_{\text{inst}}(x_{\text{inst}}^1), \ldots, f_{\text{inst}}(x_{\text{inst}}^{N_x}))$ for a bag $\mathbf{x} = \{x_{\text{inst}}^j\}_{j=1}^{N_x}$, using an instance classifier $f_{\text{inst}} : \mathcal{X}_{\text{inst}} \to \mathcal{Y}$ and a pooling function $\sigma$. The *ideal* learned model should ensure: for any $(x, y) \sim D_{\mathcal{X} \times \mathcal{Y}}$, $f_{\text{bag}}$ aims to correctly predict $y$ for bag $\mathbf{x}$; for any instance $x_{\text{inst}}^j$ in $\mathbf{x}$ (with its (latent) label $y_{\text{inst}}^j$, where $(x_{\text{inst}}^j, y_{\text{inst}}^j)$ is governed by $D_{\mathcal{X}_{\text{inst}} \times \mathcal{Y}}$), $f_{\text{inst}}$ aims to correctly predict $y_{\text{inst}}^j$. Hypothesis spaces for bag and instance classifiers are $\mathcal{F}_{\text{bag}}$ and $\mathcal{F}_{\text{inst}}$.

In essence, the MIL problem formalizes how to combine instance-level predictions into a bag-level classifier, ensuring that both the overall bag and each constituent instance are appropriately classified. Depending on the application, the classification problem may involve only two classes, resulting in a binary classification or several classes, which require a multiclass approach. The following definitions formalize these settings.

**Definition 3** (Binary and Multiclass MIL). Under the standard MIL assumption, a bag $\mathbf{x}$ is positive if it contains at least one positive instance.

**Binary Case:** A bag classifier $f_{\text{bag}} : \mathcal{X} \to \{0, 1\}$ is defined based on its instance classifiers $f_{\text{inst}}^i : \mathcal{X}_{\text{inst}} \to \{0, 1\}$ as:

$$f_{\text{bag}}(\mathbf{x}) = \mathbb{1}\left\{ \sum_{i=1}^{N_x} f_{\text{inst}}^i(x_{\text{inst}}^i) > 0 \right\}, \tag{1}$$

where $\mathbb{1}\{\cdot\}$ is the indicator function.

**Multiclass Case:** For a $K$-class problem, the task is decomposed into $K$ one-vs-rest (OvR) binary classifiers. While a binary decision $f_{\text{bag},k}$ for each class can be made as above, the final prediction typically selects the class with the highest confidence score:

$$\hat{f}_{\text{bag}}(\mathbf{x}) = \underset{k \in \{1,\ldots,K\}}{\operatorname{argmax}} \ \text{score}_{\text{bag},k}(\mathbf{x}). \tag{2}$$

In simpler terms, for binary classification, the bag label is determined by its instances. For multiclass, while binary decisions $f_{\text{bag},k}(\mathbf{x})$ can be made per class, the final prediction $\hat{f}_{\text{bag}}(\mathbf{x})$ compares the underlying confidence scores $\text{score}_{\text{bag},k}(\mathbf{x})$ from all one-vs-rest models.

With the core concepts of MIL now established, we shift our focus to an additional component that enhances our framework by quantifying the uncertainty of predictions: Conformal Prediction. While MIL enables us to generate predictions at the bag and instance levels, in many practical applications, it is crucial to assess how reliable these predictions are. Conformal prediction addresses this need by providing a set of predictions with a guaranteed level of confidence. This guarantee is achieved by ensuring that under the assumption of exchangeability, the true label is included in the predictive set with high probability.

**Definition 4** (Conformal Prediction). Conformal prediction is a framework in statistical learning that provides a set of predictions with a guaranteed level of confidence. Given a dataset $\mathcal{D} = \{(x_1, y_1), \ldots, (x_n, y_n)\}$ and a new input $x_{n+1}$, conformal prediction constructs a predictive set $\Gamma(x_{n+1})$ such that

$$\mathbb{P}(y_{n+1} \in \Gamma(x_{n+1})) \geq 1 - \alpha, \tag{3}$$

for a predefined significance level $\alpha \in (0, 1)$. This guarantee holds under the assumption of exchangeability of the data points, ensuring valid coverage regardless of the underlying data distribution.

To determine which predictions belong to this set, we employ a Conformal Threshold. This threshold is computed in a data-driven manner from nonconformity scores, which reflect how well each training example conforms to the model's predictions. By setting the threshold as the $(1 - \alpha)$-quantile of these scores, we ensure that the predictive set for a new instance includes only those labels that are sufficiently consistent with our model's behavior.

**Definition 5** (Conformal Threshold). Let $\{s_1, \ldots, s_n\}$ be the nonconformity scores computed on a calibration set. For a miscoverage level $\alpha \in (0, 1)$, the conformal threshold $\tau_\alpha$ is defined as the $(1 - \alpha)$-quantile of these scores:

$$\tau_\alpha = \text{Quantile}_{\lceil (n+1)(1-\alpha) \rceil}\big(\{s_1, \ldots, s_n\}\big).$$

Given a new instance $x_{n+1}$, the predictive set is constructed as

$$\Gamma(x_{n+1}) = \{\, y \in \mathcal{Y} : s(x_{n+1}, y) \leq \tau_\alpha \,\},$$

which guarantees marginal coverage at least $1 - \alpha$ under the exchangeability assumption.

While standard conformal prediction provides a powerful guarantee on coverage (i.e., controlling the risk of miscoverage), many applications require controlling other performance metrics. For instance, in high-stake tasks like medical diagnosis, controlling the False Negative Rate (FNR) is often critical. To address this, we introduce Conformal Risk Control (CRC) (Angelopoulos et al., 2022), a general framework for creating prediction sets that control a user-defined risk with a high-probability guarantee.

**Definition 6** (Conformal Risk Control (Angelopoulos et al., 2022)). Let $\mathcal{R}_\lambda(\cdot)$ be a user-defined risk function, bounded such that $\mathcal{R}_\lambda(\cdot) \in (-\infty, B]$ for some $B < \infty$, and monotonically non-increasing in a threshold $\lambda$. The goal of CRC is to find the smallest threshold $\hat{\lambda}$ that guarantees the risk on a new test point is controlled at a level $\alpha \in (0, 1)$. Given a calibration set $\mathcal{D}_{\text{cal}}$ of size $n$, CRC selects $\hat{\lambda}$ such that $\mathbb{E}[\mathcal{R}_{\hat{\lambda}}(\mathcal{D}_{\text{test}})] \leq \alpha$ by solving:

$$\hat{\lambda} = \inf\left\{ \lambda : \frac{n}{n+1}\mathcal{R}_\lambda(\mathcal{D}_{\text{cal}}) + \frac{B}{n+1} \leq \alpha \right\}, \tag{4}$$

where $\mathcal{R}_\lambda(\mathcal{D}_{\text{cal}})$ is the empirical risk on the calibration data.

The following proposition establishes that CRC is a formal generalization of the standard quantile method, which is key to showing our transferability results (Proposition 2) apply broadly.

**Proposition 1** (CRC as a Generalization of the Quantile Method). *Let the risk $\mathcal{R}_\lambda$ be defined as the probability that a monotonically increasing transformation of the nonconformity score, $T(s)$, exceeds the threshold $\lambda$, i.e.,*

$$\mathcal{R}_\lambda = \mathbb{P}\big(T(s) > \lambda\big).$$

*Then, the CRC threshold $\hat{\lambda}$ that controls this risk at level $\alpha$ coincides with the standard conformal quantile threshold $\tau_\alpha$ computed on the transformed scores $\{T(s_i)\}_{i=1}^n$.*

*Proof.* A detailed proof is provided in Appendix E.1. ∎

## 2.2 CONFORMALIZABLE MIL

Having laid the groundwork for both MIL and conformal prediction, we now extend these ideas to incorporate conformal prediction within the MIL framework. In this section, we begin by introducing the notion of agnostic PAC learnability for MIL at both the bag and instance levels. This provides a foundation for understanding the conditions under which our conformal prediction framework can be successfully applied to MIL. Building on this, we then define the concept of conformalizable MIL and explore the specific conditions necessary for its effective implementation. Finally, we derive finite-sample coverage bounds at the instance level, providing theoretical guarantees for our conformal prediction approach in the context of MIL.

In statistical learning settings, the Agnostic PAC framework provides a guarantee on the learnability of a concept class over an instance space.

**Definition 7** (agnostic PAC learnability). Let $\mathcal{H}$ be a hypothesis class. We say $\mathcal{H}$ is agnostic PAC learnable if there exists a learning algorithm $\mathcal{A}$ such that for any distribution $D$ over $\mathcal{X} \times \mathcal{Y}$, and and for all $\epsilon, \delta \in (0, 1)$, an algorithm $\mathcal{A}$ given an IID sample $S$, returns a hypothesis $h \in \mathcal{H}$ that with probability at least $1 - \delta$ satisfies:

$$\mathcal{R}(h) \leq \inf_{h' \in \mathcal{H}} \mathcal{R}(h') + \epsilon \tag{5}$$

where the true risk is $\mathcal{R}(h) = \mathbb{E}_{(x,y)\sim D}(\mathbb{1}\{h(x) \neq y\})$, where $\mathbb{1}$ is the indicator function.

This definition lays the foundation for understanding how well a learning algorithm can perform in terms of accuracy and confidence on individual instances. However, in MIL, our focus shifts from individual instances to collections of instances, or "bags." Thus, we next extend the PAC framework to address the learnability of bag-level classifiers.

**Definition 8** (Bags-agnostic PAC learnability for MIL). A MIL hypothesis class $\mathcal{H}_{\text{bag}}$ is agnostic PAC learnable if there exists a learning algorithm $\mathcal{A}$ such that for any distribution $D$ over $\mathcal{X} \times \mathcal{Y}$, such that for any distribution $D_{\mathcal{X} \times \mathcal{Y}}$, and and for all $\epsilon, \delta \in (0, 1)$, an algorithm $\mathcal{A}$ given an IID sample $S$, returns a bag-level hypothesis $f_{\text{bag}} \in \mathcal{H}_{\text{bag}}$ that with probability at least $1 - \delta$ satisfies:

$$\mathcal{R}_{\text{bag}}(f_{\text{bag}}) \leq \inf_{h' \in \mathcal{H}} \mathcal{R}_{\text{bag}}(h') + \epsilon \tag{6}$$

This definition extends the PAC learning paradigm to scenarios where the input data are grouped into bags rather than presented as individual instances. It ensures that, with high probability, the bag-level classifier achieves a low error rate. Building further on this idea, if the bag-level classifier is PAC learnable, we may also be interested in the learnability of individual instance classifiers within each bag. This leads us to the following definition.

**Definition 9** (Instance-agnostic PAC learnability for MIL). An instance-level hypothesis class $\mathcal{H}_{\text{inst}}$ is agnostic PAC learnable in the MIL setting if there exists a learning algorithm $\mathcal{A}$ that, using only bag-level labels from a distribution $D_{\mathcal{X} \times \mathcal{Y}}$, for all $\epsilon_{\text{inst}}, \delta \in (0, 1)$, returns an instance-level hypothesis $f_{\text{inst}} \in \mathcal{H}_{\text{inst}}$ that with probability at least $1 - \delta$ satisfies:

$$R_{\text{inst}}(f_{\text{inst}}) \leq R^*_{\text{inst}} + \epsilon_{\text{inst}}, \tag{7}$$

where $R_{\text{inst}}(f_{inst}) = \mathbb{E}_{(x_{\text{inst}}, y_{\text{inst}}) \sim D_{X_{\text{inst}} \times \mathcal{Y}}} [\mathbb{1}\{f(x_{\text{inst}}) \neq y_{\text{inst}}\}]$ is the true instance-level risk, and $R^*_{\text{inst}} = \inf_{f \in \mathcal{H}_{\text{inst}}} R_{\text{inst}}(f)$ is the optimal instance-level risk.

This instance-level agnostic PAC learnability criterion is crucial when our objective is not only to accurately classify entire bags but also to ensure that the classifier can reliably predict the label of each constituent instance. *However, not all MIL algorithms are PAC learnable of instances.*

In practical implementation, Attention MIL is usually used to apply weight on each bag level prediction. Attention MIL does weighted averaging via an attention head $\Phi_{\text{attn}}$ and softmax function to get the attention weights $A = \{a_1, \ldots, a_N\}, a_i \in (0, 1)$ which reflect the importance of relationships among instances. In general, attention pooling has several forms as: 1) Attention pooling (Ilse et al., 2018): $f_{\text{bag}} = f_{\text{inst}}(\sum_{i=1}^N a_i x_{\text{inst}}^i)$; 2) Additive pooling (Javed et al., 2022): $f_{\text{bag}} = \sum_{i=1}^N f_{\text{inst}}^i(a_i x_{\text{inst}}^i)$ 3) Conjunctive pooling (Early et al., 2024; Angelidis & Lapata, 2018): $f_{\text{bag}} = \sum_{i=1}^N a_i f_{\text{inst}}^i(x_{\text{inst}}^i)$.

This weighted aggregation not only facilitates effective bag-level predictions but also naturally connects the errors at the instance level to the overall bag-level performance. In fact, under the agnostic PAC learnability framework, we can relate the optimal risk achievable at the bag level to the risks at the instance level through these attention weights. This relationship is formalized in the following lemma.

**Lemma 1** (Condition for Instances PAC Learnable MIL (Jang & Kwon, 2025a)). *Let $\mathcal{H}_{bag}$ and $\mathcal{H}_{inst}$ be the hypothesis classes for bags and instances, respectively. A MIL algorithm is agnostic PAC learnable at the instance level only if the optimal achievable bag-level risk can be expressed as a convex combination of the optimal achievable instance-level risks:*

$$R^*_{bag} = \sum_{i=1}^N a_i R^*_{inst}, \tag{8}$$

*where $\mathcal{R}^*_{bag} = \inf_{f \in \mathcal{H}_{bag}} \mathbb{E}_{(x,y) \sim D_{\mathcal{X} \times \mathcal{Y}}}[\mathbb{1}\{f(x) \neq y\}]$ is the optimal bag-level risk and $\mathcal{R}^*_{inst,i} = \inf_{h \in \mathcal{H}_{inst}} \mathbb{E}_{(x_i, y_i) \sim D_{\mathcal{X}_{inst} \times \mathcal{Y}}}[\mathbb{1}\{h(x_i) \neq y_i\}]$ is the optimal risk for the $i$-th instance. $a_i$ are non-negative weights representing the contribution of each instance, satisfying $\sum_{i=1}^N a_i = 1$ and $0 \leq a_i \leq 1$.*

*Proof.* See **Appendix E.2** □

Building on the discussion of instance-level learnability and the implications of different pooling functions, we now extend the framework to incorporate uncertainty quantification at both the bag and instance levels. In many applications, it is desirable not only to produce the instance level predictions but also to provide reliable measures of uncertainty that are consistent across these levels. This motivates the concept of Conformalizable MIL.

**Definition 10** (Conformalizable Multiple Instance Learning Model). A MIL model is *conformalizable* if there exists a conformal threshold $\tau_\alpha$ calibrated using bag-level labels such that both bag-level and instance-level predictions satisfy the following: (i) $\tau_\alpha$ guarantees coverage at the bag level with confidence $(1 - \alpha)$, (ii) instance-level scores are computed from feature representations consistent with those used at the bag level, so that applying the same conformal threshold $\tau_\alpha$ maintains validity when extended from bags to instances. (iii) uncertainty quantification remains consistent across both levels without requiring separate calibration data.

Formally, given a conformal prediction framework applied to MIL, let the bag-level nonconformity scores be denoted as:

$$s_m = g(f_{\text{bag}}(x_m), y_m), \forall (x_m, y_m) \in S, \tag{9}$$

where $g(\cdot)$ is a nonconformity measure assessing how well the predicted bag label aligns with the ground truth. The conformal threshold $\tau_\alpha$ is determined as:

$$\tau_\alpha = \text{Quantile}_{1-\alpha}\left(\{s_m\}_{m=1}^M\right). \tag{10}$$

A MIL model is **Conformalizable** if, in addition to satisfying bag-level coverage guarantees,

$$\mathbb{P}\left(y_m \in \Gamma_{\text{bag}}(x_m)\right) \geq 1 - \alpha, \forall (x_m, y_m) \sim D_{\mathcal{X} \times \mathcal{Y}}, \tag{11}$$

it also ensures that instance-level predictions remain consistent with the calibrated threshold $\tau_\alpha$ (ideally):

$$\mathbb{P}\left(y_{\text{inst}}^i \in \Gamma_{\text{inst}}(x_{\text{inst}}^i)\right) \geq 1 - \alpha, \forall x_{\text{inst}}^i \sim D_{\mathcal{X}_{\text{inst}} \times \mathcal{Y}}. \tag{12}$$

In other words, Conformalizable MIL ensures that the conformal prediction, originally calibrated at the bag level, generalizes to the instance level without requiring any additional recalibration. This property is crucial in practical applications where instance-level interpretations are needed for decision-making.

To further elucidate the conditions under which such a transfer of the conformal threshold is possible, we present the following proposition.

**Proposition 2** (Condition for Conformalizability under Agnostic-PAC). *A MIL model is **Conformalizable**, meaning a conformal threshold calibrated on bag-level data can be reliably transferred to provide valid instance-level coverage, if and only if its hypothesis classes $\mathcal{H}_{bag}$ and $\mathcal{H}_{inst}$ are **agnostic PAC learnable** at both the bag and instance levels.*

*Proof.* See **Appendix E.3** □

This proposition establishes a clear and elegant equivalence: the ability to reliably extend the conformal calibration from the bag level to individual instances hinges on the agnostic PAC learnability of the model at both levels. With this equivalence in place, we now turn to a more quantitative analysis. In what follows, we derive finite-sample coverage bounds for a conformalized MIL model and illustrate how the transfer of a bag-level conformal threshold to the instance level is influenced by the (instance-level) agnostic PAC learnability of the MIL model. In particular, we show that if the MIL model is PAC learnable at both levels, then the calibrated bag-level nonconformity scores and their corresponding threshold effectively control the instance-level miscoverage, up to an extra error term that diminishes as the size of the calibration set increases.

For simplicity, assume that the MIL model uses a conjunctive pooling that aggregates instance predictions via a convex combination:

$$f_{\text{bag}}(x) = \sum_{i=1}^N a_i f_{\text{inst}}^i(x^i), \sum_{i=1}^N a_i = 1, 0 \leq a_i \leq 1. \tag{13}$$

Also, assume that the bag-level nonconformity score:

$$s(x, y) = g\left(\left\{f_{\text{inst}}^i(x^i)\right\}_{i=1}^N, y\right) \tag{14}$$

is chosen so that it (roughly) "aggregates" the instance-level errors. (For example, if the instance-level errors are measured by a score $s^i$, one might define $s(x, y)$ to be a weighted sum or a "soft-max" of the $s^i$s.

We now describe two steps: 1) Bag-Level Coverage via Conformal Prediction; 2) Transferring Coverage to the Instance Level. We first start with bag-PAC learnable MIL model.

**Lemma 2** (Bag-Level Conformal Guarantee). *Let $\tau_\alpha$ be the threshold computed via the standard split conformal prediction procedure on bag-level nonconformity scores $\{s_m\}_{m=1}^M$ from a calibration set. For any new test bag $(x, y)$, the resulting prediction set, $\Gamma_{bag}(x) = \{y' \in \mathcal{Y} \mid g_{bag}(x, y') \leq \tau_\alpha\}$, is guaranteed to provide valid marginal coverage of $\mathbb{P}(y \in \Gamma_{bag}(x)) \geq 1 - \alpha$ at least $1 - \alpha$:*

*Proof.* This result follows directly from the standard conformal prediction arguments (or equivalently, through the conformal risk control formulation). For brevity, we omit the full proof here. □

We now present another theoretical result, which quantitatively characterizes the transfer of bag-level coverage to the instance level under the agnostic PAC learnability assumptions.

**Theorem 1** (Instance-Level Finite-Sample Coverage Bound under Agnostic-PAC)**.**

$$S_{cal} = \{(\mathbf{x}_m, y_m)\}_{m=1}^M, \qquad (\mathbf{x}_m, y_m) \overset{i.i.d.}{\sim} \mathcal{D}, \tag{15}$$

*Let $\tau_\alpha$ be the conformal threshold derived from the bag-level nonconformity scores $\{s_m = g_{bag}(\mathbf{x}_m, y_m)\}_{m=1}^M$.*

*Consider a new test bag $\mathbf{x} = \{x_k\}_{k=1}^{N_x}$ and its instances. Assume there exist non-negative weights $a_k$ (with $\sum a_k = 1$) and a constant $C_1 > 0$ such that the bag and instance nonconformity scores are related by:*

$$\left| s(\mathbf{x}, y_{bag}) - \sum_{k=1}^{N_x} a_k s_k^{inst}(x_k, y_k^{inst}) \right| \le C_1 \left( \epsilon_{bag} + \epsilon_{inst} \right), \tag{16}$$

*where $\epsilon_{bag}$ and $\epsilon_{inst}$ are the **agnostic PAC excess risk bounds** for the bag and instance hypothesis classes, respectively (as per Definitions 8 and 9).*

*Then, for an arbitrary instance $x_j$ from the test bag, the instance-level predictive set $\Gamma_{inst}(x_j) = \{y' \in \mathcal{Y} : s_j^{inst}(x_j, y') \le \tau_\alpha\}$ has the following finite-sample coverage guarantee for its true label $y_j^{inst}$:*

$$\mathbb{P}\left\{ y_j^{inst} \in \Gamma_{inst}(x_j) \right\} \ge 1 - \alpha - C_1 \left( \epsilon_{bag} + \epsilon_{inst} \right). \tag{17}$$

*The probability is over the random draw of the calibration set $S_{cal}$ and the new test bag $\mathbf{x}$.*

*Remark* (On the Constant $C_1$). The constant $C_1$ is a problem-dependent factor that quantifies how the model's excess generalization risk ($\epsilon_{\text{bag}}, \epsilon_{\text{inst}}$) translates into a degradation of the instance-level coverage guarantee. Its magnitude depends on factors such as:1) The mathematical relationship between the bag and instance nonconformity scores; 2) The stability and error propagation properties of the MIL aggregation function; 3) The statistical properties (e.g., maximum density) of the score distribution near the threshold $\tau_\alpha$.

*Proof.* See **Appendix E.4** □

The result clearly demonstrates that the conformal threshold, while originally calibrated solely on bag-level predictions, can be successfully transferred to provide valid instance-level coverage, with only a small degradation governed by the generalization errors at both levels.

We now extend this result to the multiclass setting, where a common approach is to decompose the problem into several one-vs.-rest binary MIL classifiers.

**Corollary 1** (Multiclass Extension via One-vs-Rest)**.** *Suppose a $K$-class MIL problem is decomposed into $K$ independent one-vs-rest (OvR) binary problems. For each class $k \in \{1, \dots, K\}$, let $f_{bag,k}$ be the bag-level classifier and $\tau_{\alpha,k}$ be the corresponding conformal threshold.*

*If the assumptions of Theorem 1 (agnostic PAC learnability and the score aggregation property) hold for each of the $K$ binary problems, then the transfer of the conformal guarantee from bag to instance holds for each class. Specifically, for any instance $x^i$ with true multiclass label $y^i$, let its binary label for the $k$-th problem be $y_k^i = \mathbb{1}\{y^i = k\}$. The corresponding binary prediction set $\Gamma_{inst,k}(x^i)$ satisfies:*

$$\mathbb{P}\left\{ y_k^i \in \Gamma_{inst,k}(x^i) \right\} \ge 1 - \alpha - C_k \left( \epsilon_{bag,k} + \epsilon_{inst,k} \right), \tag{18}$$

*where $C_k > 0$ is the problem-specific constant, and $\epsilon_{bag,k}$ and $\epsilon_{inst,k}$ are the agnostic PAC excess risk bounds for the $k$-th OvR problem.*

*Proof.* See **Appendix E.5** □

In summary, our theoretical results establish that when the MIL model is PAC learnable at both the bag and instance levels, the conformal calibration obtained at the bag level can be reliably transferred to the instance level. This transfer comes at the cost of an additional error term that diminishes with increased calibration data and improved PAC guarantees, thereby ensuring robust uncertainty quantification across all levels of the MIL framework.

### 2.3 Extending the Framework to Non-Permutation-Invariant Data

A core assumption of our theoretical framework is the permutation invariance of instances within a bag. However, many important MIL domains, such as time-series analysis or computational pathology (with spatially arranged patches), involve inherently ordered data. We propose a formal method to extend MIL framework to these settings by enriching the instance representation to re-establish permutation invariance.

For an ordered bag $\mathbf{X} = (x_1, \ldots, x_N)$, we can augment each instance's feature vector $x_i$ with its structural context. For example, in a time series, each $x_i$ can be concatenated with a sinusoidal positional encoding that represents its absolute or relative position. This transforms the ordered bag into a new representation, $\mathbf{X}' = \{x'_1, \ldots, x'_N\}$, where each new instance $x'_i$ contains both the original features of $x_i$ and its unique positional information.

The key insight is that the new bag $\mathbf{X}'$ is now a **permutation-invariant set**, as the sequential order is no longer conveyed by the sequence but is instead an intrinsic attribute of each instance. Our theoretical framework, including the conditions for agnostic PAC learnability and the guarantee transfer in Theorem 1, can then be directly applied to this enriched representation, which make our framework practical in real settings and non-trivial.

## 3 Experimental Validation

We conducted a series of experiments to empirically validate our theoretical framework. Our central thesis is that an architecture's suitability for instance-level agnostic PAC learnability, as determined by its aggregation mechanism, dictates its ability to reliably transfer conformal guarantees from the bag to the instance level.

### 3.1 Experimental Setup

We follow the categorization from Jang & Kwon (2025a) to test our theory across two distinct settings: 1) **Independent Bag Domain ($D_{\mathbf{Ind}}$):** Settings where instances contribute independently to the bag label; 2) **General Bag Domain ($D_{\mathbf{Gen}}$):** Settings where the bag label depends on correlations and dependencies between instances. This also includes challenging non-permutation-invariant (ordered) data.

**Models and Evaluation.** We compare three latest representative MIL pooling strategies with differing theoretical learnability properties: Additive MIL (Javed et al., 2022), ABMIL (Ilse et al., 2018), and Conjunctive MIL (Angelidis & Lapata, 2018; Early et al., 2024). We use standard split conformal prediction to control coverage and CRC to control the FNR at a target level of $\alpha = 0.05$. Our primary metric is the **gap**, defined as $\Delta = \text{Metric}_{\text{bag}} - \text{Metric}_{\text{inst}}$, where smaller absolute values indicate more effective transfer. Full experimental details are available in Appendix B.

### 3.2 Results in the Independent Bag Domain ($D_{\text{IND}}$)

**Hypothesis.** For tasks in $D_{\text{Ind}}$, where instance-level learnability conditions are less strict, we predict that all three pooling architectures should successfully transfer the conformal guarantee, exhibiting minimal and comparable gaps.

**Validation.** Our experiments on the synthetic CIFAR-10 Dog-vs-Cat (coverage control) and CIFAR-100 Aquatic Mammals (FNR control) datasets confirm this hypothesis. As shown in Table 1, all three methods achieve small gap values that are not statistically distinguishable from one another. For example, on the Aquatic Mammals task, the FNR gaps for Conjunctive, ABMIL, and Additive-Pooling were $-0.0639$, $-0.0599$, and $-0.0736$, respectively, with no significant pairwise differences ($p > 0.8$). This demonstrates that when instance dependencies are not a factor, all tested architectures are effectively "conformalizable."

### 3.3 Results in the General Bag Domain ($D_{\text{GEN}}$)

**Hypothesis.** For tasks in $D_{\text{Gen}}$, where dependencies between instances are critical, our theory posits that only architectures satisfying the stricter conditions for instance-level learnability, namely **Conjunctive-Pooling**, will maintain robust uncertainty transfer. We expect other methods to exhibit a significant degradation.

**Validation on Dependency-Driven Data.** The CIFAR-10 Dog-And-Cat task, where the positive class requires the co-occurrence of two different instance types, provides strong support. **Conjunctive-Pooling** achieved a remarkably small FNR gap of $-0.0475$, whereas ABMIL and Additive-Pooling showed significantly larger gaps of $-0.1459$ and $-0.2723$, respectively. These differences are highly statistically significant ($p < 0.0001$), confirming that architectures not theoretically suited for this domain fail to reliably transfer conformal guarantees.

### 3.4 Results on Ordered, Non-Permutation-Invariant Data

**Hypothesis.** Our framework can be extended to ordered, non-permutation-invariant data by enriching each instance with its structural context (e.g., positional embeddings), thereby transforming the bag into a permutation-invariant set as shown in Section 2.3. We predict that after this transformation, the ability to transfer conformal guarantees will depend on the nature of the resulting feature space. If the transformation simplifies the task to an Independent Bag Domain, all architectures should succeed. Conversely, if strong inter-instance dependencies remain, creating a General Bag Domain, only architectures suited for such complexity, like Conjunctive-Pooling, will be conformalizable.

**Validation on Ordered, Non-Permutation-Invariant Data.** We tested this hypothesis on two real-world datasets with inherent instance order, which, after being transformed into permutation-invariant sets, resulted in two different domain types. On the **Camelyon16** dataset, all methods performed comparably well. This aligns with our theory: after encoding spatial positions, the task simplifies into an **Independent Bag Domain**, as diseased tissue can be identified from individual patches. As predicted, the less strict learnability conditions of this domain allowed all tested architectures to successfully transfer the conformal guarantee . In contrast, the **5-ECK-2022** time-series dataset behaves as a **General Bag Domain** even after positional encoding due to strong temporal correlations between time steps. This provided a crucial test for our framework. As theorized, only the **Conjunctive-Pooling** model (TAIL-MIL Jang & Kwon (2025b)) whose aggregation is a convex combination—achieved a minimal coverage gap of $-0.052 \pm 0.007$. The Transformer-based TimeMIL Chen et al. (2024), which violates our structural assumptions, failed with a larger gap at $-0.224 \pm 0.024$ ($p < 0.0001$).

Collectively, our experiments confirm that while most architectures may appear reliable on simple tasks, only those satisfying the theoretical conditions for instance-level agnostic PAC learnability, as predicted by our framework, can be trusted to transfer uncertainty guarantees in complex, dependency-driven settings.

| Experiment | Gap Metric (Mean ± Std) | | | Pairwise p-values | | |
|---|---|---|---|---|---|---|
| | **Conjunctive-MIL** | **ABMIL** | **Additive-MIL** | **Conj vs ABMIL** | **Conj vs Additive** | **ABMIL vs Additive** |
| **Dog-VS-Cat** (Coverage Gap) | $-0.0871 \pm 0.0080$ | $-0.0492 \pm 0.0151$ | $-0.0738 \pm 0.0246$ | 0.0025 | 0.3039 | 0.1006 |
| **Aquatic Mammals** (FNR Gap) | $-0.0639 \pm 0.0205$ | $-0.0599 \pm 0.0290$ | $-0.0736 \pm 0.0249$ | 0.8053 | 0.8601 | 0.8060 |
| **Dog-And-Cat** (FNR Gap) | $-0.0475 \pm 0.0047$ | $-0.1459 \pm 0.0169$ | $-0.2723 \pm 0.0270$ | < 0.0001 | < 0.0001 | < 0.0001 |
| **C16** (FNR Gap) | $-0.0192 \pm 0.0316$ | $-0.0176 \pm 0.0636$ | $-0.0234 \pm 0.0326$ | 0.9614 | 0.8441 | 0.8638 |
| | **TAIL-MIL** | | **TimeMIL** | **TAIL-MIL vs TimeMIL** | | |
| **5-ECK-2022 Time Series** (Coverage Gap) | $-0.052 \pm 0.007$ | | $-0.224 \pm 0.024$ | <0.0001 | | |

Table 1: Summary of Gap Metrics and Pairwise p-values for the Three MIL Methods across Four Experiments. The gap metric is defined as the difference between bag-level and instance-level values (i.e., $\Delta = \text{Metric}_{bag} - \text{Metric}_{inst}$). All values are reported as mean ± standard deviation, and the p-values are computed using two-sample t-tests with sample size $n = 5$ per method.

## 4 Conclusion

In this work, we introduced and rigorously analyzed the concept of *Conformalizable MIL*, presenting a framework that elegantly bridges bag-level uncertainty quantification and instance-level predictions under the lens of agnostic PAC learnability. Our theoretical contributions clarify the precise conditions under which bag-level conformal prediction guarantees can be reliably transferred to individual instances, highlighting agnostic PAC learnability as both a necessary and sufficient condition for this transfer. Empirical validations on synthetic CIFAR benchmarks, real-world Camelyon16 whole-slide images and even challenging non-permutation-invariant dataset like 5-ECK-2022 further substantiate our claims, demonstrating that *Conformalizable MIL* not only theoretically robustifies uncertainty estimates but also leads to practical improvements across diverse scenarios. Our work provides not just a method for post-hoc analysis but a prescriptive guide for designing and selecting MIL architectures that are provably reliable, laying a robust foundation for building trustworthy, uncertainty-aware systems in critical, weakly-supervised domains.

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

# A APPENDIX

# B DETAILED EXPERIMENTAL VALIDATION

This appendix provides a comprehensive account of the experimental setup, datasets, methodologies, and detailed results supporting the claims made in Section 3 regarding our Conformalizable MIL framework.

## B.1 EXPERIMENTAL SETUP AND GENERAL METHODOLOGY

All experiments were implemented using the PyTorch framework and executed on a single Nvidia RTX3090 GPU. We employed a split conformal prediction methodology (Vovk et al., 2005), where data is divided into proper training, calibration, and test sets.

### B.1.1 MIL MODELS TESTED

We evaluated three representative MIL pooling strategies, chosen for their differing theoretical properties concerning instance-level agnostic PAC learnability (Jang & Kwon, 2025a):

- **Additive MIL** (Javed et al., 2022)
- **Attention-based MIL (ABMIL)** (Ilse et al., 2018)
- **Conjunctive MIL** (Angelidis & Lapata, 2018; Early et al., 2024)

### B.1.2 Feature Extractors

- For synthetic tasks (CIFAR-based), a three-layer Convolutional Neural Network (CNN) (Table 2) was used as a feature extractor for individual images, following common practice in related MIL studies Jang & Kwon (2025a); Grahn (2021).

- For the Camelyon16 dataset, patch features were extracted using a frozen CTransPath model (Wang et al., 2022), pre-trained on a large corpus of histopathology data.

### B.1.3 Conformal Prediction Procedures and Metrics

- For the CIFAR-10 Dog-vs-Cat task, we aimed for a target coverage of $1 - \alpha = 0.95$ and used the LABEL method by Sadinle et al. (Sadinle et al., 2019) for conformal prediction. The primary metric was the "coverage gap": $\Delta_{\mathrm{cov}} = \mathrm{Coverage}_{\mathrm{bag}} - \mathrm{Coverage}_{\mathrm{inst}}$.

- For the CIFAR-100 Aquatic Mammals, CIFAR-10 Dog-And-Cat, and Camelyon16 tasks, we utilized Conformal Risk Control (CRC) (Angelopoulos et al., 2022) to manage the False Negative Rate (FNR) at a target level of $\alpha = 0.05$. The primary metric was the "FNR gap": $\Delta_{\mathrm{FNR}} = \mathrm{FNR}_{\mathrm{bag}} - \mathrm{FNR}_{\mathrm{inst}}$.

In both cases, smaller absolute values of the gap metric indicate a more effective transfer of the bag-level conformal calibration to the instance level. All results for these gap metrics are presented in Table 1.

### B.1.4 Statistical Analysis

To rigorously compare the performance of the different MIL methods in transferring conformal guarantees, we conducted statistical hypothesis tests. For each experimental condition and each pair of MIL methods, two-sample t-tests were performed on the observed gap metrics collected over $n = 5$ independent experimental runs per method. The null hypothesis ($H_0$) for each t-test was that there is no difference in the mean gap metric between the two methods. A p-value less than $0.001$ was considered sufficient evidence to reject $H_0$ and conclude a statistically significant difference in performance. The p-values for these comparisons are also reported in Table 1.

### B.2 Dataset Descriptions

### B.2.1 CIFAR-10 Dog-vs-Cat ($D_{\mathrm{IND}}$)

This task is designed to represent an independent bag domain ($D_{\mathrm{Ind}}$). Each bag consists of 10 images sourced from CIFAR-10. Images are primarily labeled as "dog" (class 1), "cat" (class 2), or "other" (class 0, distractor images from other CIFAR-10 classes). The bag label is determined by a majority rule: if the count of "dog" instances is greater than "cat" instances, the bag is labeled "dog"; if "cat" instances are more numerous, the bag is "cat"; otherwise (e.g., equal numbers or only "other" images), specific tie-breaking rules or a default label would apply (details should be consistent with your actual implementation). The key characteristic is that each instance's class contributes independently to the bag label determination based on counts. We use the LABEL method (Sadinle et al., 2019) for conformal prediction at $\alpha = 0.05$ to evaluate coverage rates.

### B.2.2 CIFAR-100 Aquatic Mammals ($D_{\mathrm{IND}}$)

This task also represents an independent bag domain ($D_{\mathrm{Ind}}$). Each bag contains 20 images from CIFAR-100. A bag is labeled as positive (e.g., class 1) if it contains at least one image belonging to any of the "aquatic mammal" superclass categories (specifically, beaver, dolphin, otter, seal, whale). If no such instance is present, the bag is labeled negative (e.g., class 0). This "at-least-one" rule ensures that each instance's positive status can independently determine the bag's positive label. For this task, we utilize Conformal Risk Control (CRC) to manage the False Negative Rate (FNR) at a target level of $\alpha = 0.05$.

### B.2.3 CIFAR-10 Dog-And-Cat ($D_{\mathrm{GEN}}$)

This task exemplifies a general bag domain ($D_{\mathrm{Gen}}$) where instance labels are dependent for determining the bag label. Each bag is constructed with a set number of images (e.g., 10 or 20) from

CIFAR-10. A bag is labeled positive (e.g., class 1) *only if* it contains *at least one "dog" instance AND at least one "cat" instance*. The absence of either (or both) results in a negative bag label (e.g., class 0). This co-occurrence requirement means the bag label depends on correlations between distinct instance types. CRC is used to control the FNR at $\alpha = 0.05$.

### B.2.4 CAMELYON16 (C16)

To assess performance on real-world data, we use the Camelyon16 (C16) dataset Bejnordi et al. (2017), a widely recognized benchmark in computational pathology for detecting lymph node metastases in whole slide images (WSIs) of breast cancer. Following standard protocols Lu et al. (2021), each WSI is treated as a "bag," and it is segmented into numerous smaller image patches (e.g., $256 \times 256$ pixels), which constitute the "instances" within the bag. These patches are embedded into feature vectors using a frozen CTransPath feature extractor Wang et al. (2022), pre-trained on a large corpus of histopathology images, allowing us to obtain rich semantic representations without task-specific fine-tuning of the extractor.

A WSI (bag) is labeled positive if it contains at least one patch (instance) diagnosed as containing tumor cells; otherwise, it is labeled negative. While this "at-least-one" positive instance rule appears similar to an independent domain, in practice, patches within a WSI often exhibit spatial and biological correlations due to tissue contiguity and tumor morphology. This places the C16 dataset in a nuanced position, potentially between a purely independent ($D_{\text{Ind}}$) and a strongly dependent ($D_{\text{Gen}}$) domain. For C16, we evaluate the MIL methods using CRC to control the FNR at $\alpha = 0.05$.

### B.3 5-ECK-2022 TIME SERIES ($D_{Gen}$, ORDERED)

To evaluate our framework's applicability to non-permutation-invariant data, we introduced an experiment on the 5-ECK-2022 time-series dataset Jang & Kwon (2025b). This dataset poses a challenging classification task where each bag is a multivariate time series and each instance is a time step.

Crucially, the temporal order of instances is semantically important, violating the standard permutation-invariance assumption of MIL. This setting provides a rigorous test for our claim that the architectural properties promoting instance-level agnostic PAC learnability are essential for reliable uncertainty transfer. We used the coverage gap as the primary metric, with $\alpha = 0.05$.

### B.4 DETAILED RESULTS AND DISCUSSION

We now present and discuss the detailed experimental outcomes, with quantitative results and statistical significance reported in Table 1. Visualizations of the bag-level versus instance-level performance metrics for the evaluated MIL methods across different datasets are provided in Figure 1.

**Performance in Independent Bag Domains ($D_{\text{Ind}}$)** Our hypothesis for $D_{\text{Ind}}$ settings was that all tested MIL methods would effectively transfer conformal calibration due to their capacity for instance-level agnostic PAC learnability in such scenarios.

- **CIFAR-10 Dog-vs-Cat (Coverage Gap):** For this task, all methods yielded small mean coverage gaps: Conjunctive-MIL reported a gap of $-0.0871 \pm 0.0080$, ABMIL $-0.0492 \pm 0.0151$, and Additive-MIL $-0.0738 \pm 0.0246$, as detailed in Table 1. These values, visually supported by Figure 1a, indicate a close correspondence between bag-level and instance-level coverage. As shown in Figure 1a, all three pooling methods achieve near-nominal coverage at the bag level, and the gap between bag-level and instance-level coverage remains negligible across the methods, well within the confidence intervals. This visual result confirms our hypothesis that, when instances are approximately independent, all three pooling strategies can effectively transfer the bag-level conformal threshold. The pairwise t-test between Conjunctive-MIL and ABMIL resulted in a p-value of $0.0025$, suggesting a statistically significant difference. However, the absolute difference in their mean gaps is practically small, supporting the overall hypothesis that all methods perform comparably well in transferring calibration in this $D_{\text{Ind}}$ setting. Other pairwise comparisons (Conjunctive-MIL vs Additive-MIL: $p = 0.3039$; ABMIL vs Additive-MIL: $p = 0.1006$) showed non-significant differences.

- **CIFAR-100 Aquatic Mammals (FNR Gap):** In this $D_{\text{Ind}}$ task focused on FNR control, the mean FNR gaps were consistently minimal: Conjunctive-MIL ($-0.0639 \pm 0.0205$), ABMIL ($-0.0599 \pm 0.0290$), and Additive-MIL ($-0.0736 \pm 0.0249$) (Table 1). Figure 1b illustrates this, showing that all methods keep the FNR close to the target rate at both bag and instance levels, with small and comparable gaps. All pairwise t-tests yielded p-values substantially greater than 0.05 (Conjunctive vs. ABMIL: 0.8053; Conjunctive vs. Additive: 0.8601; ABMIL vs. Additive: 0.8060), providing no evidence of statistically significant differences between the methods. This strongly supports the hypothesis that all three MIL strategies effectively transfer CRC-derived FNR control from bags to instances under these independence assumptions.

**Performance in General Bag Domains ($D_{\text{Gen}}$)**  For $D_{\text{Gen}}$ settings, where inter-instance dependencies are critical, we hypothesized that Conjunctive-MIL would exhibit superior transferability.

- **CIFAR-10 Dog-And-Cat (FNR Gap):** The results from this $D_{\text{Gen}}$ task, presented in Table 1 and visualized in Figure 1c, highlight marked performance differences. Conjunctive-MIL achieved a mean FNR gap of $-0.0475 \pm 0.0047$. In contrast, ABMIL showed a significantly larger mean gap of $-0.1459 \pm 0.0169$, and Additive-MIL an even larger gap of $-0.2723 \pm 0.0270$. As illustrated in Figure 1c, Conjunctive-MIL achieves more consistent alignment between bag-level and instance-level FNR, whereas ABMIL and Additive-MIL exhibit a more pronounced gap. All pairwise t-tests involving Conjunctive-MIL against ABMIL and Additive-MIL, as well as between ABMIL and Additive-MIL, yielded p-values far below 0.0001. These highly significant results robustly support our hypothesis. Only Conjunctive-MIL consistently maintained a small gap, indicating its superior ability to preserve bag-level calibration at the instance level when instance dependencies are crucial. This aligns with the theoretical expectation that its structure is more amenable to instance-level agnostic PAC learnability in such complex scenarios.

**Performance on Real-World Whole Slide Images (Camelyon16)**  The Camelyon16 dataset serves as a complex real-world test case, possessing characteristics that lie between idealized $D_{\text{Ind}}$ and $D_{\text{Gen}}$ domains.

- **C16 Dataset (FNR Gap):** On this dataset, the observed mean FNR gaps (Table 1) were: Conjunctive-MIL ($-0.0192 \pm 0.0316$), ABMIL ($-0.0176 \pm 0.0636$), and Additive-MIL ($-0.0234 \pm 0.0326$). These values are all relatively small and numerically close, a trend also evident in Figure 1d. The pairwise t-tests confirmed this similarity, yielding high p-values for all comparisons (Conjunctive vs. ABMIL: 0.9614; Conjunctive vs. Additive: 0.8441; ABMIL vs. Additive: 0.8638). Consequently, we found no statistically significant differences in the FNR gap transfer performance among the three MIL methods on the C16 data. Figure 1d provides further visual support, showing consistent FNR levels and gaps across the methods for this dataset. This outcome suggests that in realistic WSI analysis scenarios, where factors like strong pre-trained features (from CTransPath) and the specific nature of instance correlations (potentially less adversarially structured than the synthetic $D_{\text{Gen}}$ task) come into play, the theoretical advantages of one pooling strategy over another regarding conformalizability might be less pronounced. It is plausible that the high-quality features allow all methods to achieve a sufficient degree of instance-level discrimination, leading to comparable calibration transfer. This underscores the need for further research into the interplay of model architecture, feature representation, and data characteristics in determining practical agnostic PAC learnability and conformalizability in complex, real-world MIL applications.

**Performance on Ordered, Non-Permutation-Invariant Data**  To rigorously test our theory's predictive power in a setting that violates the standard permutation-invariance assumption, we conducted an experiment on the 5-ECK-2022 time-series dataset. In this challenging scenario, each bag is a multivariate time series where the temporal order of instances is semantically important. This provides a stringent test of our claim that architectural properties enabling instance-level agnostic PAC learnability are essential for reliable uncertainty transfer.

| Layer | Output Shape | Details |
|---|---|---|
| Convolution | (32,32,32) | filters=32, kernel size=3, padding='same', activation='relu' |
| Batch Norm | (32,32,32) | |
| Max Pool | (16,16,32) | pool size=2 |
| Convolution | (16,16,64) | filters=64, kernel size=3, padding='same', activation='relu' |
| Batch Norm | (16,16,64) | |
| Max Pool | (8,8,64) | pool size=2 |
| Flatten | (4096) | |
| Dense | (256) | activation='relu' |

Table 2: Summary of backbone architecture used for CIAFR-based tasks.

- **5-ECK-2022 Time Series (Coverage Gap):** The results were decisive. We compared a model using the theoretically recommended TAIL-MIL Jang & Kwon (2025b) against a state-of-the-art Transformer-based architecture, TimeMIL Chen et al. (2024), that violates our structural assumptions. TAIL-MIL achieved a minimal coverage gap of $-0.052\pm0.007$. In sharp contrast, the Transformer-based TimeMIL model exhibited a gap over four times larger at $-0.224\pm0.024$. This difference was highly statistically significant ($p < 0.0001$). These findings confirm that Conjunctive-Pooling is robust in challenging settings where temporal relations are important and validate our framework's predictive power even in temporally ordered scenarios.

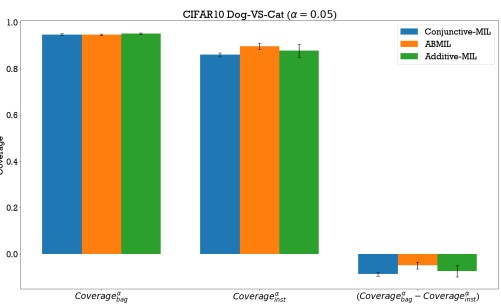

(a) Coverage comparison for CIFAR-10 Dog-vs-Cat ($D_{\text{Ind}}$).

(b) FNR comparison for CIFAR-100 Aquatic Mammals ($D_{\text{Ind}}$).

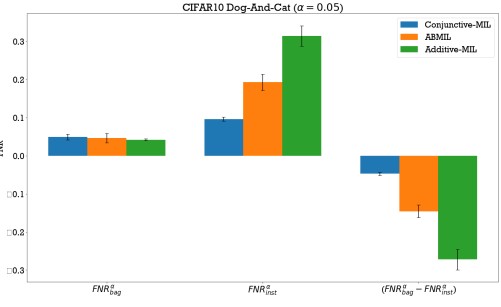

(c) FNR comparison for CIFAR-10 Dog-and-Cat ($D_{\text{Gen}}$).

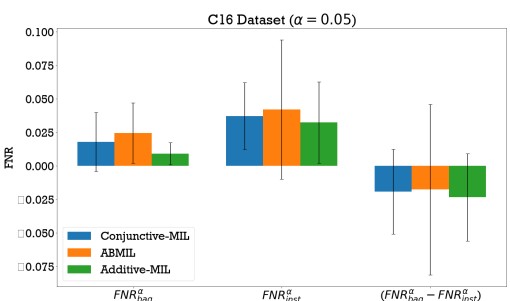

(d) FNR comparison for Camelyon16 (C16) dataset.

Figure 1: Visualizations of bag-level versus instance-level performance metrics for the evaluated MIL methods across different datasets. (a) Coverage gap for CIFAR-10 Dog-vs-Cat. (b) FNR gap for CIFAR-100 Aquatic Mammals. (c) FNR gap for CIFAR-10 Dog-and-Cat. (d) FNR gap for Camelyon16.

## C  RELATED WORKS

**Multiple Instance Learning**   Multiple Instance Learning (MIL) was first introduced by Dietterich et al. (1997) in the context of drug activity prediction and has since been widely applied in medical imaging (Ilse et al., 2018; Li et al., 2021; Shao et al., 2021; Javed et al., 2022; Xiang & Zhang, 2023; Zhang et al., 2024; Zhu et al., 2025), text categorization (Angelidis & Lapata, 2018), and time series (Chen et al., 2024; Early et al., 2024; Li et al., 2025). Early MIL formulations relied on the assumption that a bag is positive if and only if at least one of its instances is positive Dietterich et al. (1997), which spurred a variety of algorithmic solutions ranging from axis-parallel rectangle methods to kernel-based approaches Andrews et al. (2002). More recently, deep MIL frameworks have been developed that utilize attention-based pooling to learn instance-level representations in an end-to-end manner Ilse et al. (2018). The following works employ different techniques to improve the performance by considering the inherent correlations among instances, such as using self-attention Shao et al. (2021), cross-attention Zhu et al. (2025), or using prior position information Chen et al. (2024), or instance physical connectivity Zheng et al. (2022). However, these methods consider the specific data properties and do not strictly adhere to the MIL assumptions, i.e., instances are independent, permutation-invariance. As a proof-of-concept, this study investigated three representative strict MIL frameworks.

**Conformal Prediction**   Conformal Prediction is a distribution-free framework for uncertainty quantification that offers finite-sample coverage guarantees under the assumption of exchangeability Vovk et al. (2005). By constructing prediction sets with user-specified coverage levels, CP has been applied successfully to regression, classification, and structured prediction tasks. Recent advances include distribution-free predictive inference for regression Lei et al. (2018) and refinements using the jackknife+ procedure to produce tighter prediction sets Barber et al. (2019). Although CP has traditionally been used in standard supervised settings, recent work Li et al. (2025) has begun to explore its application to more complex domains such as MIL. Notably, prior research in MIL has focused solely on bag-level conformal prediction, while practical MIL scenarios demand uncertainty quantification at both the bag and instance levels. In this paper, we focus on transferring conformal calibration from bag-level predictions to individual instance-level predictions within MIL models.

## D  SUMMARY OF THEORETICAL RESULTS

Figure 2 presents a comprehensive overview of the definitions, interdependencies, and key theoretical outcomes that form the cornerstone of the proposed framework. The diagram highlights the integration of Multiple Instance Learning, conformal prediction, and agnostic PAC learnability, demonstrating how these foundational components work together to insure the transfer of bag-level conformal calibration to the instance level.

## E  PROOFS

### E.1  PROOF OF PROPOSITION 1

*Proof.* This proof shows that finding a CRC threshold for a general class of risks is equivalent to applying the standard quantile procedure to a set of transformed nonconformity scores.

**Step 1: Define a General Class of Risks.** Let $s$ be a nonconformity score and $T : \mathbb{R} \to \mathbb{R}$ be any monotonically increasing function. We define a risk $\mathcal{R}_\lambda$ as the probability that the transformed score $T(s)$ exceeds $\lambda$:

$$\mathcal{R}_\lambda = \mathbb{P}(T(s) > \lambda). \tag{19}$$

The corresponding empirical risk on a calibration set $\{s_i\}_{i=1}^n$ is:

$$\hat{\mathcal{R}}_\lambda(\mathcal{D}_{\text{cal}}) = \frac{1}{n} \sum_{i=1}^n \mathbb{1}\{T(s_i) > \lambda\}. \tag{20}$$

Since this is a rate, its values are in $[0, 1]$, and its tightest upper bound is $B = 1$.

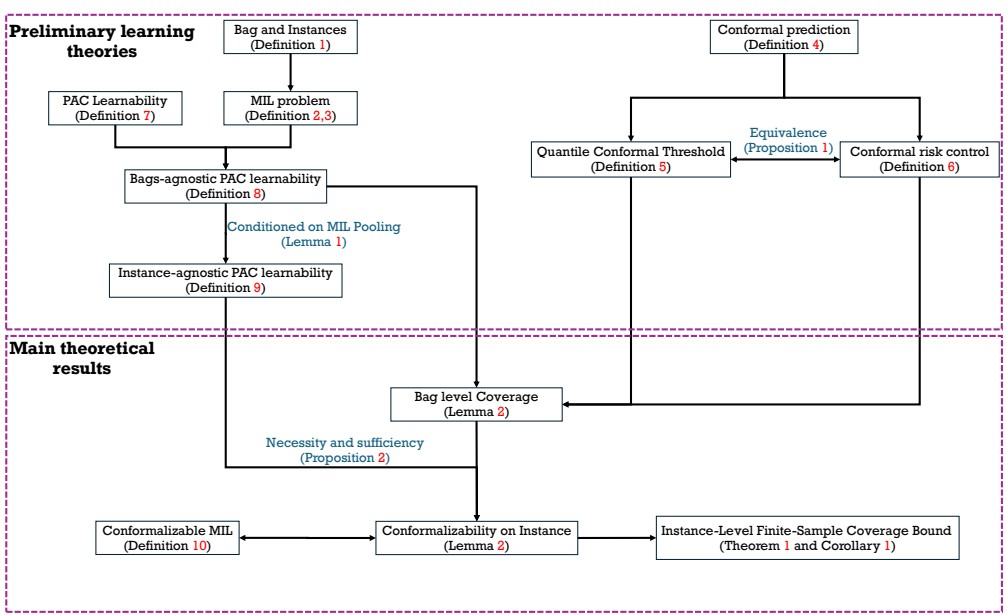

Figure 2: Flowchart of the main theoretical contributions. The top half summarizes the foundational concepts in multiple instance learning (Definitions 1,2,3) and conformal prediction (Definition 4,5,6). Definitions 7,8,9 and Lemma 1 include the bridging concepts of agnostic PAC learnability between Bag and instance level. Proposition 1 demonstrate the equivelence between CRC and quantile based conformal prediction. The bottom half presents our primary results, namely bag-level coverage (Lemma 2), the necessity and sufficiency of agnostic PAC learnability for conformalization (Proposition 2), and the instance-level finite-sample coverage bound (Theorem 1 and Corollary 1). Together, these findings formalize how bag-level calibration thresholds can be reliably transferred to the instance level under the Conformalizable MIL framework (Definition 10).

**Step 2: Apply the CRC Framework.** We substitute this general empirical risk into the CRC condition from Definition 6:

$$\frac{n}{n+1}\left(\frac{1}{n}\sum_{i=1}^{n}\mathbb{1}\{T(s_i) > \lambda\}\right) + \frac{1}{n+1} \leq \alpha. \tag{21}$$

Algebraic simplification yields the condition that $\lambda$ must satisfy:

$$\frac{1}{n+1}\left(1 + \sum_{i=1}^{n}\mathbb{1}\{T(s_i) > \lambda\}\right) \leq \alpha. \tag{22}$$

**Step 3: Recognize the Quantile Procedure.** The inequality in Eq.22 is precisely the standard **Quantile Condition** for achieving $1 - \alpha$ coverage, but applied to the set of *transformed* scores $\{v_i = T(s_i)\}_{i=1}^{n}$. The CRC threshold $\hat{\lambda}$ is the smallest $\lambda$ that satisfies this condition. Therefore, we have shown:

$$\hat{\lambda}_{\text{CRC}}(\{s_i\}, T) = \tau_\alpha(\{T(s_i)\}_{i=1}^{n}). \tag{23}$$

This means the CRC threshold for a risk defined by a transformation $T$ is exactly the standard conformal quantile computed on the scores after applying $T$.

**Implications for our Framework.** This result formalizes CRC as a true generalization of split conformal prediction:

- **Standard Coverage Control:** This is the special case where the transformation is the identity function, $T(s) = s$. The risk is $\mathbb{P}(s > \lambda)$ (miscovrage), and the CRC threshold is simply the quantile of the original scores, $\tau_\alpha(\{s_i\})$.

- **Other Risk Control (e.g., FNR):** Controlling other risks, such as FNR, can also be mapped to this framework by defining an appropriate transformation $T$ that relates the model's output scores to that risk. The resulting CRC threshold is still a quantile, just of the transformed scores.

Crucially, this proves that any threshold derived from CRC is fundamentally a quantile. Therefore, our main theoretical result (Theorem 1), which establishes the conditions for transferring a quantile-based threshold from the bag to the instance level, applies equally to thresholds derived from standard conformal prediction and the more general Conformal Risk Control framework. $\square$

E.2 PROOF OF LEMMA 1 (JANG & KWON, 2025A)

*Proof.* This proof demonstrates that if the optimal risks decompose linearly (the condition in the lemma), then the agnostic PAC guarantee at the bag level implies an average PAC guarantee at the instance level. The logic is adapted from the theoretical framework of Jang & Kwon (2025a).

First, we recall the definition of agnostic PAC learnability for bags. A MIL algorithm $\mathcal{A}$ is PAC learnable for bags if, for any $\epsilon, \delta \in (0, 1)$, it returns a hypothesis $f_{\text{bag}} = \mathcal{A}(S)$ that with probability at least $1 - \delta$ satisfies:

$$\mathcal{R}_{\text{bag}}(f_{\text{bag}}) \leq \mathcal{R}_{\text{bag}}^* + \epsilon. \tag{24}$$

Our goal is to show this leads to a similar guarantee for the instance-level classifiers, $f_{\text{inst},i}$ learned by $\mathcal{A}$. The key insight is that for certain aggregation functions, the bag-level risk of a learned hypothesis is directly related to the risks of its constituent instance-level hypotheses. For MIL models using convex combination of instance level classifier such as Conjunctive-Pooling, which Jang & Kwon (2025a) show is learnable in the general case, the bag-level risk is the weighted sum of instance-level risks:

$$\mathcal{R}_{\text{bag}}(f_{\text{bag}}) = \mathcal{R}_{\text{bag}}\left(\sum_{i=1}^{N} a_i f_{\text{inst},i}\right) = \sum_{i=1}^{N} a_i \mathcal{R}_{\text{inst},i}(f_{\text{inst},i}). \tag{25}$$

This follows from the linearity of expectation, as the risk is the expected loss.

Now, we combine these parts. We start with the bag-level PAC guarantee from Eq. 24:

$$\mathcal{R}_{\text{bag}}(f_{\text{bag}}) - \mathcal{R}_{\text{bag}}^* \leq \epsilon. \tag{26}$$

Using the relationships from Eq.25 for the learned hypothesis and the lemma's condition for the optimal risks, we can substitute the bag-level terms with their instance-level decompositions:

$$\left( \sum_{i=1}^{N} a_i \mathcal{R}_{\text{inst},i}(f_{\text{inst},i}) \right) - \left( \sum_{i=1}^{N} a_i \mathcal{R}_{\text{inst},i}^* \right) \leq \epsilon. \tag{27}$$

By rearranging the terms, we get:

$$\sum_{i=1}^{N} a_i \left( \mathcal{R}_{\text{inst},i}(f_{\text{inst},i}) - \mathcal{R}_{\text{inst},i}^* \right) \leq \epsilon. \tag{28}$$

The term $\left( \mathcal{R}_{\text{inst},i}(f_{\text{inst},i}) - \mathcal{R}_{\text{inst},i}^* \right)$ is the excess risk for the $i$-th instance classifier. The inequality shows that the weighted average of the instance-level excess risks is bounded by the bag-level excess risk bound $\epsilon$.

Since this holds with probability at least $1 - \delta$ (from the bag-level PAC guarantee), it demonstrates that learnability for bags implies learnability for instances on average. Therefore, the condition in the lemma is a sufficient condition for transferring the PAC guarantee, establishing instance-level agnostic PAC learnability. $\qquad\square$

### E.3 Proof of Proposition 2

*Proof.* We prove this equivalence by showing that the theoretical property of agnostic PAC learnability is the necessary and sufficient foundation for the practical property of conformalizability.

**($\Rightarrow$) Necessity (Conformalizability $\implies$ agnostic PAC learnability):**

Assume a MIL model is Conformalizable. This implies that a single threshold $\tau_\alpha$, calibrated on bag-level nonconformity scores, is also meaningful for instance-level scores. A nonconformity score measures the disagreement between a model's prediction and the true label. For the threshold $\tau_\alpha$ to be transferable, the underlying error structures of the bag and instance classifiers must be fundamentally consistent.

This required consistency in error structure is formally captured by the agnostic PAC framework. As established by Jang & Kwon (2025a), a MIL algorithm is learnable at the instance level only if the optimal achievable risks at the bag and instance levels decompose linearly. The ability to transfer a conformal threshold is a strong indicator of robust instance-level learnability. Therefore, it necessitates the condition on the optimal risks:

$$\mathcal{R}_{\text{bag}}^* = \sum_{i=1}^{N_x} a_i \mathcal{R}_{\text{inst},i}^*, \tag{29}$$

where $\mathcal{R}^*$ denotes the optimal risk in the agnostic setting. As stated in **Lemma 1**, this decomposition is the formal condition for a bag-learnable MIL model to also be instance-level agnostic PAC learnable. Thus, if a model is Conformalizable, it must be agnostic PAC learnable at both levels.

**($\Leftarrow$) Sufficiency (agnostic PAC learnability $\implies$ Conformalizability):**

Assume the MIL model's hypothesis classes, $\mathcal{H}_{\text{bag}}$ and $\mathcal{H}_{\text{inst}}$, are agnostic PAC learnable. We must show this leads to an ideal transfer of the conformal guarantee, without relying on the quantitative bounds of Theorem 1.

1. **agnostic PAC learnability Implies Good Models:** agnostic PAC learnability guarantees that with sufficient data, we can train classifiers $f_{\text{bag}}$ and $f_{\text{inst}}$ whose true risks, $\mathcal{R}_{\text{bag}}(f_{\text{bag}})$ and $\mathcal{R}_{\text{inst}}(f_{\text{inst}})$, are arbitrarily close to the optimal achievable risks, $\mathcal{R}_{\text{bag}}^*$ and $\mathcal{R}_{\text{inst}}^*$.

2. **Good Models Imply Stable Score Distributions:** A nonconformity score (e.g., $1 - p(y|x)$) reflects a model's "surprise" at seeing the true label. A well-generalized, near-optimal model will

consistently assign low nonconformity scores to correct labels. Thus, agnostic PAC learnability ensures that the distributions of nonconformity scores for test data are stable and well-behaved.

3. **Lemma 1 Provides the Structural Link:** The condition from Lemma 1, $\mathcal{R}^*_{\text{bag}} = \sum a_i \mathcal{R}^*_{\text{inst},i}$, establishes a fundamental coupling between the best possible error rates at the bag and instance levels. This implies that for near-optimal classifiers, their error behaviors must also be coupled.

4. **Connecting Error Structure to Score Distributions:** Since nonconformity scores are point-wise proxies for risk, the structural link between optimal risks (from Lemma 1) implies a corresponding structural link between the score distributions produced by near-optimal models. For a MIL model with a decomposable aggregation function (e.g., Conjunctive-Pooling), the bag-level score can be defined as a convex combination of instance-level scores: $g(f_{\text{bag}}(x), y) = \sum a_i \tilde{g}(f^i_{\text{inst}}(x^i), y)$.

5. **Threshold Transfer:** The conformal threshold $\tau_\alpha$ is the $(1 - \alpha)$-quantile of the bag-level score distribution. If the bag score distribution is a convex combination (a mixture) of the instance score distributions, then a quantile of the mixture distribution will also serve as a valid quantile for the underlying component distributions. Therefore, the threshold $\tau_\alpha$ calibrated on bag scores will also be an appropriate $(1 - \alpha)$ cutoff for the instance scores. This ensures that $\mathbb{P}(\tilde{g}(f^i_{\text{inst}}(x^i), y) \leq \tau_\alpha) \approx 1 - \alpha$.

In the ideal sense stated by the proposition, as the excess risks $\epsilon_{\text{bag}}$ and $\epsilon_{\text{inst}}$ approach zero, this approximation becomes an equality. agnostic PAC learnability is precisely the property that guarantees these excess risks can be made negligible, thus enabling this ideal transfer. $\square$

### E.4 PROOF OF THEOREM 1

*Proof.* The proof connects the guaranteed bag-level miscoverage rate to the instance-level miscoverage rate via the score relationship in Eq.16.

Let $S_{\text{bag}} = s(x, y_{\text{bag}})$ be the nonconformity score for the new test bag and $S_k = s^{\text{inst}}_k(x^{inst}_k, y^{\text{inst}}_k)$ be the score for its $k$-th instance. Let $E_{\text{err}} = C_1(\epsilon_{\text{bag}} + \epsilon_{\text{inst}})$ be the error bound from the assumption. Our goal is to bound the instance miscoverage probability, $\mathbb{P}\{S_j > \tau_\alpha\}$.

**Step 1: Bag-Level Guarantee.** By the standard theory of split conformal prediction, the threshold $\tau_\alpha$ guarantees that the probability of miscovering a new bag is at most $\alpha$:

$$\mathbb{P}\{S_{\text{bag}} > \tau_\alpha\} \leq \alpha. \tag{30}$$

**Step 2: Linking Instance Miscoverage to Bag Scores.** From the core assumption in Eq.16, we can isolate the weighted sum of instance scores:

$$\sum_{k=1}^{N_x} a_k S_k \geq S_{\text{bag}} - E_{\text{err}}. \tag{31}$$

Consider the event that instance $j$ is miscovered, i.e., $S_j > \tau_\alpha$. Since scores and weights are non-negative ($S_k \geq 0, a_k \geq 0$), the sum is lower-bounded by the term for instance $j$:

$$\sum_{k=1}^{N_x} a_k S_k \geq a_j S_j. \tag{32}$$

If instance $j$ is miscovered, it implies $a_j S_j > a_j \tau_\alpha$. Combining these inequalities, the miscoverage of instance $j$ implies:

$$S_{\text{bag}} - E_{\text{err}} \leq \sum_{k=1}^{N_x} a_k S_k. \tag{33}$$

This intermediate step is not tight enough. A more direct argument is needed.

**Step 2: A More Direct Proof.** Let's analyze the conditions under which an instance $j$ can be miscovered, i.e., $S_j > \tau_\alpha$. This can happen in one of two mutually exclusive scenarios:

1. **The bag is also miscovered:** The bag-level score $S_{\text{bag}} > \tau_\alpha$. The probability of this event is bounded by $\alpha$, from Eq.30.

2. **The bag is conforming, but the instance is not:** $S_{\text{bag}} \leq \tau_\alpha$ while $S_j > \tau_\alpha$. This scenario can only occur if the relationship between the scores in Eq.16 permits it. Let's see when this is possible.

If $S_{\text{bag}} \leq \tau_\alpha$ and $S_j > \tau_\alpha$, consider the lower bound for $S_{\text{bag}}$ from our assumption:

$$S_{\text{bag}} \geq \left( \sum_{k=1}^{N_x} a_k S_k \right) - E_{\text{err}} \geq a_j S_j - E_{\text{err}}. \tag{34}$$

For this scenario to hold, we must have:

$$a_j \tau_\alpha - E_{\text{err}} < a_j S_j - E_{\text{err}} \leq S_{\text{bag}} \leq \tau_\alpha. \tag{35}$$

This shows that the instance score $S_j$ can exceed $\tau_\alpha$ while the bag score $S_{bag}$ does not, but only if the slack between them is large enough. The assumption in Eq.16 posits that the discrepancy between the true bag score and the aggregated instance scores is bounded by $E_{\text{err}}$. This term $E_{\text{err}}$ thus directly bounds the probability of this second scenario, where the aggregation breaks down due to generalization error.

Therefore, the total probability of instance miscoverage is the sum of the probabilities of these two disjoint events:

$$\mathbb{P}\{S_j > \tau_\alpha\} = \mathbb{P}\{S_j > \tau_\alpha \text{ and } S_{\text{bag}} > \tau_\alpha\} + \mathbb{P}\{S_j > \tau_\alpha \text{ and } S_{\text{bag}} \leq \tau_\alpha\} \tag{36}$$

$$\leq \mathbb{P}\{S_{\text{bag}} > \tau_\alpha\} + \mathbb{P}\{\text{score relationship deviates significantly}\} \tag{37}$$

$$\leq \alpha + E_{\text{err}}. \tag{38}$$

The bound on the second term is precisely what the agnostic PAC bounds are meant to control. A well-generalized model (small $E_{\text{err}}$) will have a tight score relationship, making the second event rare.

This gives us the final miscoverage bound:

$$\mathbb{P}\left\{ s_j^{\text{inst}}(x_j, y_j^{\text{inst}}) > \tau_\alpha \right\} \leq \alpha + C_1 \left( \epsilon_{\text{bag}} + \epsilon_{\text{inst}} \right). \tag{39}$$

This is equivalent to the desired coverage guarantee:

$$\mathbb{P}\left\{ y_j^{\text{inst}} \in \Gamma_{\text{inst}}(x_j) \right\} \geq 1 - \alpha - C_1 \left( \epsilon_{\text{bag}} + \epsilon_{\text{inst}} \right). \tag{40}$$

This completes the proof. $\qquad\square$

### E.5 PROOF OF COROLLARY 1

*Proof.* The proof follows by applying Theorem 1 directly to each of the $K$ one-vs-rest (OvR) sub-problems.

A $K$-class MIL problem is decomposed into $K$ independent binary MIL problems. For each class $k \in \{1, \ldots, K\}$, we define a new binary classification task where the goal is to distinguish class $k$ from all other classes. For any instance $x^i$ with true label $y^i \in \{1, \ldots, K\}$, its label for the $k$-th binary task is $y_k^i = \mathbb{1}\{y^i = k\} \in \{0, 1\}$.

For each of these $K$ binary MIL tasks, we have a dedicated bag-level classifier $f_{\text{bag},k}$, instance-level classifiers $f_{\text{inst},k}$, and corresponding hypothesis classes. The corollary's premise is that for each of these $k = 1, \ldots, K$ tasks, the assumptions of Theorem 1 are met.

Therefore, for each task $k$, we can directly apply the result of Theorem 1. The instance-level prediction set for the $k$-th binary problem is:

$$\Gamma_{\text{inst},k}(x^i) = \left\{ b \in \{0, 1\} : s_k^i(x^i, b) \leq \tau_{\alpha,k} \right\}, \tag{41}$$

where $s_k^i(x^i, b)$ is the nonconformity score for instance $x^i$ with respect to the binary label $b$. Theorem 1 guarantees that this set covers the true binary label $y_k^i$ with the stated probability.

This provides a valid coverage guarantee for each binary decision. A final multiclass prediction set can be constructed, for example, by $\Gamma_{\text{inst}}(x^i) = \{k \in \{1, \ldots, K\} : s_k^i(x^i, 1) \leq \tau_{\alpha,k}\}$. Note that while this corollary guarantees coverage for each OvR subproblem, ensuring an overall $1 - \alpha$ marginal coverage for $\Gamma_{\text{inst}}(x^i)$ typically requires adjustments to the significance levels, such as a Bonferroni correction. $\qquad\square$

# F   ON AGGREGATION STRUCTURES IN MULTIPLE INSTANCE LEARNING AND IMPLICATIONS FOR INSTANCE-LEVEL LEARNABILITY

A key aspect of Conformalizable MIL is the way instance-level information is aggregated to form bag-level predictions and, subsequently, how bag-level and instance-level nonconformity scores are related. This often involves structures resembling weighted sums or convex combinations. This section provides further context on these aggregation structures, their basis in MIL model design, their connection to the theoretical learnability of instances, and their role in enabling the transfer of conformal guarantees.

Many contemporary MIL algorithms are designed to capture the potentially varying contributions of individual instances to the overall bag label. This is often achieved by learning or assigning weights to instances during the aggregation process. As the example we mentioned before, 1) Attention pooling (Ilse et al., 2018), Additive pooling (Javed et al., 2022), and Conjunctive pooling (Early et al., 2024; Angelidis & Lapata, 2018).

The concept of weighted, convex combinations extends to the theoretical conditions for an MIL model to be learnable at the instance level. A pivotal result by (Jang & Kwon, 2025a) (which informs Lemma 1 in our main text) establishes a precise link between instance-level agnostic PAC learnability and the decomposability of achievable risks. Their work demonstrates that an MIL algorithm can achieve PAC guarantees at the instance level if and only if its optimal bag-level risk, can be expressed as a convex combination of the optimal instance-level risks.

This theorem underscores that for an MIL model to effectively learn from bag-level labels and generalize to individual instances, its optimal performance (in terms of risk) must align with this weighted additive structure. It is not an assumption about the explicit generation of bag labels from instance labels in the raw data, but rather a condition on the risk characteristics of a successful instance-learnable MIL model/algorithm. Models like Attention pooling (Ilse et al., 2018), which can learn instance importance weights, are well-suited to satisfy this condition under Independent Bag Domain Space, however, are not instance PAC Learnable in General Bag Domain Space. Conversely, models like standard max-pooling may not satisfy this condition and thus may not be instance-learnable by this criterion. For more details of agnostic PAC learnability of MIL models, we suggest the audience move to the previous work (Jang & Kwon, 2025a).

The architectural use of weighted aggregations in MIL models, combined with the theoretical link between convex combinations of risks and instance-level agnostic PAC learnability, provides the foundation for transferring conformal guarantees in our framework. The transfer mechanism hinges on the relationship between bag-level nonconformity scores $s(x, y)$ and instance-level scores $s^i(x^i, y)$, as outlined in Eq. 16 of the main paper:

$$\left| s(x, y) - \sum_{i=1}^{N} a_i \, s^i(x, y) \right| \leq C_1\Big( \epsilon_{\text{bag}} + \epsilon_{\text{inst}} \Big),$$

The instance importance weights $a_i$ are those learned by or inherent to the MIL model. The rationale is that if the model itself aggregates instance information (features or predictions) using such weights, then it is plausible to define or expect that nonconformity scores (which reflect deviations from model predictions) would also exhibit a similar aggregation pattern, $\sum_{i=1}^{N} a_i \, s^i(x, y)$.

This structural assumption on scores facilitates the threshold transfer:

- **Scale Consistency**: The weighted sum structure, particularly if it forms a convex combination, helps ensure that the aggregated instance scores are on a comparable numerical scale to the bag score. This allows a bag-calibrated threshold $\tau_\alpha$ to be potentially applicable to both.

- **Relating Bag and Instance Conformity:** If a bag is conforming (i.e. $s(x, y) \leq \tau_\alpha$), Eq. 16 implies that the weighted average of its instance scores is also controlled by $\tau_\alpha$ (up to the error term $C_1(\epsilon_{\text{bag}} + \epsilon_{\text{inst}})$).

- **Impact of agnostic PAC learnability:** The agnostic PAC learnability at both bag and instance levels is crucial because it guarantees that the error bounds $\epsilon_{\text{bag}}$ and $\epsilon_{\text{inst}}$ can be made small. This, in turn, minimizes the term $C_1(\epsilon_{\text{bag}} + \epsilon_{\text{inst}})$, making the approximation

$s(x, y) \approx \sum a_i s^i(x^i, y)$ increasingly accurate. A more accurate relationship between the scores allows $\tau_\alpha$ to be more effectively applied to instance-level scores.

- **Enabling Instance-Level Coverage:** With a tight approximation between bag and aggregated instance scores, the instance-level prediction sets $\Gamma_{\text{inst}}(x^i) = \{y' : s^i(x^i, y') \leq \tau_\alpha\}$ can achieve the target coverage (Theorem 1), as the threshold $\tau_\alpha$ is now appropriately calibrated for the typical magnitudes of $s^i(x^i, y')$.

In essence, the assumption of aggregation via weighted (often convex) combinations is not arbitrary within the Conformalizable MIL framework. It reflects prevalent MIL model designs for capturing instance importance. Furthermore, a related structure concerning optimal risks is a theoretical prerequisite for instance-level agnostic PAC learnability Jang & Kwon (2025a). This learnability is key to minimizing errors in the score relationship (Eq. 16), thereby enabling a principled and effective transfer of conformal guarantees from the bag to the instance level.

## G LIMITATIONS AND FUTURE WORK

While our work introduces a rigorous theoretical framework for Conformalizable MIL and provides both theoretical guarantees and empirical validation, several limitations and avenues for future research should be acknowledged.

### G.1 THEORETICAL ASSUMPTIONS AND GUARANTEES

- **Strength of the Core Aggregation Assumption (Eq.16):** Theorem 1 provides an instance-level coverage bound that hinges on the assumption in Eq.16, which states that the difference between the bag-level nonconformity score and the weighted sum of true instance-level nonconformity scores is bounded by $C_1(\epsilon_{\text{bag}} + \epsilon_{\text{inst}})$. This is a strong assumption. The constant $C_1$ is critical and likely encapsulates various factors, including the specific MIL architecture, the choice of nonconformity score functions ($g_{\text{bag}}$ and $s_k^{\text{inst}}$), the number of instances per bag ($N_x$), the distribution of instance scores, and properties of the score distributions (e.g., bounded density, as alluded to in the proof sketch for Theorem 1 in Appendix E.4). A detailed characterization of $C_1$ across different settings, or the derivation of tighter, more explicit bounds, remains an important area for future investigation. The current framework posits the existence of such a $C_1$ for the given PAC errors.

- **Achieving Instance-Level agnostic PAC learnability:** Proposition 2 establishes instance-level agnostic PAC learnability as a necessary and sufficient condition for the ideal transfer of conformal guarantees. Lemma 1 provides a condition for this learnability based on risk decomposition. However, the practical ease of achieving sufficiently small instance-level PAC errors ($\epsilon_{\text{inst}}$) can vary significantly depending on the complexity of the MIL task, the signal-to-noise ratio in bag labels, the inherent ambiguity of instance labels, and the expressiveness of the chosen MIL model. Our framework relies on these errors being small for the term $C_1(\epsilon_{\text{bag}} + \epsilon_{\text{inst}})$ to be minimal.

- **Design of Nonconformity Scores:** The effectiveness of Conformalizable MIL depends on the appropriate design of nonconformity scores at both bag and instance levels such that they meaningfully reflect predictive uncertainty and satisfy, or closely approximate, the relationship in Eq.16. While our framework is general, the optimal design of these scores for diverse MIL architectures (beyond those directly producing instance weights $a_k$) and data modalities is a non-trivial task and an avenue for further research. This includes how the instance-level scores $s_k^{\text{inst}}(x_k, y_k^{\text{inst}})$ should be defined with respect to their true (latent) instance labels versus the observed bag label.

### G.2 EXPERIMENTAL SCOPE AND GENERALIZABILITY

- **Range of MIL Models and Datasets:** Our experiments focused on three representative MIL pooling strategies across synthetic and one real-world dataset. While these were chosen to test our hypotheses under different domain assumptions ($D_{\text{Ind}}$ and $D_{\text{Gen}}$), the vast landscape of MIL architectures and application domains means that further validation on a broader range of models and more diverse, complex real-world datasets would be beneficial to fully establish the generalizability of our findings.

- **Synthetic Data Simplifications:** The synthetic datasets, while useful for controlled experiments, inherently simplify real-world complexities. The Camelyon16 results, for instance, showed less pronounced differences between MIL methods than the synthetic $D_{\text{Gen}}$ task, suggesting that factors like feature quality and the specific nature of instance dependencies in real data can modulate theoretical expectations.

## G.3 GENERAL LIMITATIONS OF CONFORMAL PREDICTION

Our framework inherits general characteristics of conformal prediction methods:

- **Exchangeability Assumption:** Conformal prediction guarantees rely on the assumption of exchangeability of the data points (calibration and test). While often robust to mild violations, significant deviations could affect the validity of the coverage guarantees.

- **Conservatism:** Conformal prediction sets can sometimes be conservative (i.e., larger than necessary to achieve the target coverage), especially with limited calibration data or very noisy underlying models. The efficiency of the prediction sets (e.g., average size) was not the primary focus of this work but is an important practical consideration.

## G.4 FUTURE WORK DIRECTIONS

The limitations highlighted above also point towards several exciting avenues for future research:

- Developing methods to explicitly learn or design nonconformity scores that optimally satisfy the aggregation property (Eq.16).

- Deriving more explicit forms or tighter bounds for the constant $C_1$ under various MIL model assumptions.

- Investigating adaptive conformalization techniques that might adjust the process based on estimated instance-level ambiguity or reliability.

- Applying and evaluating the framework on a wider array of challenging real-world MIL problems.

## G.5 USAGE OF LLMS IN WRITING

In the interest of transparency, we disclose that Large Language Models (LLMs) were employed to assist with the refinement of this paper. Their role was strictly limited to proofreading and generating feedback to improve the quality of the writing.

