# OpenReview forum: "Bridging Bag-Level and Instance-Level Uncertainty with Conformalizable MIL"
_ICLR.cc/2026/Conference — Submitted to ICLR 2026_

### Official Review · Reviewer_fTLs · 2025-10-26

**Soundness:** 3
**Presentation:** 3
**Contribution:** 3
**Rating:** 6
**Confidence:** 4

**Summary:**

This paper addresses the reliable quantification of instance-level uncertainty in Multiple Instance Learning (MIL). The authors introduce Conformalizable MIL, a novel theoretical and empirical framework that successfully bridges practical uncertainty guarantees with theoretical learnability, building on the existing work (Jang and Kwon et al., 2025), which has shown instance-level learnability in MIL.

The core contribution is a rigorous proof that instance-level Agnostic PAC learnability is both necessary and sufficient for the successful transfer of a bag-level calibrated conformal threshold to the instance level. Furthermore, the paper provides a quantitative finite-sample coverage bound, demonstrating that instance-level coverage degrades gracefully with the model's (PAC) excess generalization risk.

The authors hypothesize that an architecture's aggregation mechanism dictates its PAC learnability, which in turn governs its ability to transfer uncertainty guarantees. By testing in controlled settings, they show that only architecture theoretically suited for D_Gen successfully transfers uncertainty guarantees on complex tasks.

**Strengths:**

S1. The paper tackles a problem of high practical importance. While MIL is powerful in low-label regimes its deployment in high-stakes applications has been hindered by the lack of reliable instance-level uncertainty. This paper rigorously defines the conditions under which instance-level conformal guarantees can be achieved using only weak, bag-level supervision.

S2. The link established between the practical property of conformalizability (Definition 10) and the theoretical property of Agnostic PAC learnability (Definition 9) is a core contribution. Proposition 2 (necessity and sufficiency) and Theorem 1 (finite-sample bound) represent an integration of learning theory and conformal prediction.

S3. The formal extension of the framework to ordered non-permutation-invariant data (Section 2.3) by enriching instance representations significantly broadens the practical applicability of this work to important domains like time-series analysis.

**Weaknesses:**

W1. The work heavily builds upon the instance-level PAC learnability results for MIL from Jang & Kwon (2025a). While appropriately cited, the core condition (Lemma 1) determining instance PAC learnability is taken from this prior work.

W2. The proof of Theorem 1 relies on the assumption (Eq. 16) that the bag score is approximately a weighted sum of instance scores, with the deviation bounded by PAC errors. While plausible for certain architectures (like Conjunctive-Pooling with specific scores), its general validity across all MIL models and nonconformity scores needs careful consideration. The appendix (Sec F) provides some justification.

W3. The finite-sample bound in Theorem 1 depends on a constant $C_1$, which encapsulates how excess risk translates to coverage degradation. The paper acknowledges that $C_1$ depends on several factors but doesn't provide a method to estimate it, making the quantitative bound less practical for direct application without further analysis.

W4. The theory predicts Conjunctive MIL should excel in D_Gen domains, which holds for synthetic data but not clearly for Camelyon16. On Camelyon16, all three methods performed comparably with no statistically significant differences (p > 0.84), showing FNR gaps of -0.0192±0.0316 (Conjunctive), -0.0176±0.0636 (ABMIL), and -0.0234±0.0326 (Additive).

W5. Limited guidance on designing appropriate nonconformity scores for new domains. The paper also doesn't provide clear guidance on selecting appropriate pooling strategies for new applications.

W6. While the chosen architectures (Additive, ABMIL, Conjunctive) are representative regarding the learnability theory, many other modern MIL methods exist (e.g., TransMIL, attention mechanisms beyond ABMIL). Testing the framework on a wider range of SOTA MIL models could further strengthen the conclusions.

**Questions:**

Q1. Theorem 1's bound depends on the constant $C_1$. Can the authors provide more intuition or perhaps bounds on $C_1$ for specific common MIL architectures (e.g., Conjunctive-Pooling) and nonconformity scores (e.g., $1-\hat{p}(y|x)$)? How does $C_1$ scale with factors like bag size $N_x$?

Q2. The core assumption (Eq. 16 / Appendix F) relates bag scores to a weighted sum of instance scores. How well does this assumption hold empirically for the different MIL architectures tested? Could deviations from this assumption explain why even Conjunctive-Pooling sometimes shows a non-zero gap in practice?

Q3. Why do you think the theoretical predictions don't clearly hold on Camelyon16?

Q4. How would your framework apply to recent Transformer-based MIL methods that don't use explicit convex combinations?

---

### Official Review · Reviewer_G6MS · 2025-10-28

**Soundness:** 1
**Presentation:** 2
**Contribution:** 2
**Rating:** 0
**Confidence:** 4

**Summary:**

The paper addresses conformal prediction for multi-instance learning setup. Specifically the authors discuss the transferability of the conformal threshold computed on bag level score to the instance level score. They derive bound for the coverage guarantee thorugh PAC agnostic learnability.

**Strengths:**

I believe the problem that the authors discuss is very interesting. The approach towards that (using the PAC agnostic trainability) is a promising direction.

**Weaknesses:**

**First and foremost: the guarantee.** The authors show that the coverage guarantee for the instance level prediction sets using the quantile evaluated on the bag level samples has a coverage guarantee of 1 - alpha - some constant C. This is as general as saying that the coverage guarantee for the instance level is some 1 - beta where the beta is task dependent. The main point of conformal prediction is to provide the prediction sets with a solid guarantee that is known in prior to using the prediction set. The main interesting challenge of this setup is how to use the exchangeability of the bags to derive some exchangeability for the scores, which basically I do not see the paper showing that.

In summary the paper does not show any method to compute the constant C prior to observing the prediction set, the authors do not indicate when the assumptions like PAC learnability is even true.

**Major flaw in Definition 7.** In the original definition of the agnostic PAC learnability there is a minimum number of available samples (dataset size) called sample complexity which is a function of the variables epsilon and delta. It is not mentioned in the definition which basically reduces the definition to the case where the training algorithm should always return minimum possible error. Basically without that the definition is highly restrictive to the best possible likelihood simply by setting both delta and epsilon to zero which is valid; the resulting assumption is indeed unrealistic.

The same issue is consistently present in all three two successor definitions.

**Unclear definition of conformalizable MIL model.** The sentence “ instance-level scores are computed from feature representations consistent …” in Definition 10, is not clearly stated. I am assuming that the authors are mentioning that there is a mapping from the bag-level score function to the instance level score function that the same threshold preserves the same coverage. But I am not sure if that is exactly what the authors aim to state.

**Minor points on style and typing.** I suggest not to use MIL (short form in the title). This would be ambiguous to whom not familiar with the abbreviation.

A sentence like “providing reliable uncertainty quantification” in the abstract (although mostly used) is misleading. This is since CP is a statistical post-hoc method that is not directly connected to other uncertainty quantification techniques.

Minor typos: repeated “and” at the end of line 195, repeated use of the word “where” in line 200,

**Questions:**

Please see the weaknesses.

---

> ### Comment · Reviewer_G6MS · 2025-11-27
>
> I think before every other concern I would pose these two questions.
>
> 1. What is the lower bound on the coverage. Is the coverage probability computable without seeing the future test points? "1 - \alpha - some constant" is only acceptable if there is a procedure to bound the constant prior to observing the test point.
>
> 2. What is the role of "minimum number of available samples (dataset size)  - a.k.a. sample complexity" in \epsilon and \delta. Without this variable the definition is very restrictive -- reducing to the ground truth classifier.

---

### Official Review · Reviewer_L8J2 · 2025-10-30

**Soundness:** 2
**Presentation:** 2
**Contribution:** 2
**Rating:** 4
**Confidence:** 4

**Summary:**

This paper investigates the conditions under which bag-level conformal predictions can be transferred to the instance-level to provide reliable instance-level uncertainty estimates in MIL (multi-instance learning). Built upon the theory of instance-level learnability, the authors conduct theoretical analyses to infer the conditions. The results show that the conformal calibration can be transferred to instance-level when an MIL model is PAC learnable at both bag- and instance-level. The experiments on several datasets confirm the effectiveness of Conformalizable MIL, i.e., the framework can guide the choice of those MIL algorithms that can transfer bag-level conformal calibration to instance-level.

**Strengths:**

- This paper studies a new problem in MIL: under which conditions the bag-level conformal calibration can be transferred to the instance-level to provide reliable instance-level uncertainty estimates.
- Preliminaries are properly introduced. Moreover, they are relatively thorough. These serve as the foundations of the presented theoretical analysis.
- Theoretical analyses are conducted to infer the conditions and these may provide a guideline for selecting suitable MIL algorithms to provide reliable instance-level predictions.

**Weaknesses:**

- Important evaluations are not presented: This paper investigates the condition of transferring bag-level conformal calibration to instance-level. The experiments use the performance gap between bag and instance levels as the primary metric for evaluating the goodness of the transfer. However, they fail to demonstrate whether the instance-level prediction is good enough in terms of uncertainty estimation, while the authors claim "we provide a guide for designing and selecting MIL architectures that are not only accurate but also yield trustworthy, calibrated uncertainty estimates at the fine-grained instance level". In other words, knowing the goodness of transfer is actually not enough, since no one cares about the transfer (gap) performance if the MIL algorithm cannot even give a good coverage at the bag-level or instance-level in conformal prediction. We may care about the gap performance only if we believe the bag-level uncertainty estimates are good enough, right? Yet, in fact, this is not the case in MIL because there seems to be no literature that has shown that the conformal prediction in MIL could produce good uncertainty estimates.
- More MIL variants should be added to the analysis: In the presented three MIL algorithms, only ABMIL is commonly used and frequently adopted in weakly-supervised scenarios. The authors are encouraged to add more variants to experiments, e.g. TransMIL or DSMIL, etc.

**Questions:**

- Could the authors provide the results on test samples, instead of only the gap metrics? I am curious if the proposed comformalizable MIL could also provide reliable predictions (or accurate uncertainty estimates as mentioned by the authors in Introduction). From my understanding, this is very important, since no one cares about the transfer (gap) performance if the MIL algorithm cannot even give a good coverage at the bag-level or instance-level in conformal prediction.
- Could more MIL algorithms be added to the comparison, especially those frequently used in weakly-supervised scenarios (like TransMIL, DSMIL, or conventional MIL algorithms like mi-Net)?

---

### Meta-Review · Area_Chair_m62x · 2026-01-09

**Summary:**

This paper considers the problem of conformal prediction in a multiple-instance learning setting. It studies the conditions under which bag-level conformal prediction sets can be transferred to the instance-level and derives a bound for the coverage guarantee based on PAC agnostic learnability.

All the reviewers' pointed out a number of issues with the theoretical analysis and the underlying assumptions. Additionally, important experimental evaluations are missing to demonstrate whether the instance-level prediction set is good enough in terms of uncertainty estimation, which is important for the paper's claims.

Authors' have not responded to these critical questions. Therefore, I recommend to reject this paper and encourage the authors' to address them for re-submission to a future venue.

**Reviewer Scores:**

N/A

---

### Decision · Program_Chairs · 2026-01-26

Reject